# Multiobjective Scheduling of Hybrid Renewable Energy System Using Equilibrium Optimization

**Salil Madhav Dubey** [1], **Hari Mohan Dubey** [2], **Manjaree Pandit** [1] **and Surender Reddy Salkuti** [3,*]

1   Department of Electrical Engineering, Madhav Institute of Technology & Science, Gwalior 474005, India; salil.dubey3107@gmail.com (S.M.D.); manjaree_p@hotmail.com (M.P.)
2   Department of Electrical Engineering, Birsa Institute of Technology, Dhanbad 828123, India; hmdubey.ee@bitsindri.ac.in
3   Department of Railroad and Electrical Engineering, Woosong University, Daejeon 34606, Korea
*   Correspondence: surender@wsu.ac.kr; Tel.: +82-10-9674-1985

**Abstract:** Due to increasing concern over global warming, the penetration of renewable energy in power systems is increasing day by day. Gencos that traditionally focused only on maximizing their profit in the competitive market are now also focusing on operation with the minimum pollution level. The paper proposes a multiobjective model capable of finding a set of trade-off solutions for the joint optimization problem, considering the cost of reserve and curtailment of power from renewable sources. Managing a hybrid power system is a challenging task due to its stochastic nature mixed with the objective function and complex practical constraints associated with it. A novel metaheuristic Equilibrium Optimizer (EO) algorithm incepted in the year 2020 utilizes the concept of control volume and mass balance for finding equilibrium state is proposed here for computing the optimal schedule and impact of renewable energy integration on profit and emission for different optimization objectives. In this paper, EO has shown dominant performance over well-established metaheuristic algorithms such as particle swarm optimizer (PSO) and artificial bee colony (ABC). In addition, EO produces well-distributed Pareto-optimal solutions and the fuzzy min-ranking is used as a decision maker to acquire the best compromise solution.

**Keywords:** multi-objective; renewable energy; profit-based scheduling; Equilibrium Optimizer





## 1. Introduction

The electricity demand is increasing day by day due to the growth and evolution of industrial establishments and changing lifestyles. A substantial part of the demand is still being met by thermal power generation, which depends on fossil fuels such as coal, natural gas and petroleum, which are considered the main sources of harmful emissions and air pollution. The burning of fossil fuels releases harmful gases into the atmosphere. Globally, the power generation sector contributes more than 30% of carbon dioxide emissions to the atmosphere [1]. These pollutant gases affect not only humans but are also responsible for the destruction of other lifeforms. Due to growing concern over environmental considerations, there is a demand for sufficient and secured electricity at the lowest price along with a minimum level of pollution to stabilize the environment. It is possible by multi-objective optimization that considers power generation cost and emission both for minimization. Economic emission dispatch (EED) is a key optimization problem of the power system. The objective is to schedule the committed generator optimally, so that generation cost and emission are minimized simultaneously while satisfying all operational constraints associated with it [2]. Various solution approaches have been reported for the EED problem, broadly categorized into classical, metaheuristic and hybrid approaches [3]. As a practical EED problem, is a highly nonlinear, complex constrained optimization problem, and it is not easy to find the optimal solution to such a problem by classical methods. The metaheuristic approach can overcome the difficulties, but its

computational time is more. Metaheuristic and hybrid approach includes evolutionary algorithm (EA) [2], genetic algorithm (GA) [4], non-dominating sorting genetic algorithm (NSGA) [5–7], particle swarm optimization (PSO) [8–10], harmony search (HS) [11,12], differential evolution (DE) [13], hybrid bat algorithm (HBA) [14], kernel search optimization (KSO) [15], time-varying acceleration coefficient particle swarm optimization (TVAC-PSO) with exchange market algorithm (EMA) [16], interior search optimization (ISO) [17], grass shopper optimization (GSO) [18], sine cosine algorithm (SCA) [19], hybrid bacterial foraging with Nelder-mead [20] and many more. A detailed review metaheuristics approach for the solution of the EED problem can be found in references [21,22].

Creating new efficient power plants or identifying and expanding existing ones that produce low emission is time-consuming and require significant capital investment to fulfill the ever-increasing power demand. Another way is to integrate renewable energy resources (RER) such as wind and solar power into the existing grid. However, the integration of RER in the existing power grid creates several operational issues due to the unpredictable nature of wind speed and solar irradiation. Therefore, fluctuation in the wind and solar power generation needs more consideration. Further load demand is a random variable, and we cannot predict it accurately. The system operator can anticipate the uncertainty associated with wind power generation/solar power generation/load demand using the forecast. Generally, the probability distribution function is used to modeling the uncertainty related to RER integration. Consequently, integration of RER complicates the formulation of the EED problem significantly [21].

Simultaneous minimization of cost, emission and loss for the wind-thermal system with complex operational constraints such as valve point loading (VPL) effect, ramp rate limits (RRL), prohibited operating zones (POZ) and spinning reserve (SR) are available in reference [23]. Here, the time-varying fuzzy selection mechanism is used to rank the conflicting objective. Wind-based combined economic emission dispatch (WCEED) has been investigated in reference [24] to acquire Pareto optimal solution. Here piecewise linear approximation method is used to model wind power. The multi-area dynamic economic emission dispatch (MADEED) of the complex system comprises a cascaded hydro system, uncertain wind power and thermal generator system investigated in reference [25]. Here Weibull pdf is used for wind power calculation.

CEED of a grid comprising of wind and PV generation systems is presented in [26]. Here, a linear relationship between day-ahead forecasted output power by wind and PV system with operation and maintenance cost is used to model the objective function. Optimal generation scheduling of a hybrid generation system comprised of thermal, wind and solar has been investigated in reference [27]. Here Weibull pdf and bimodal distribution are separately used to handle the uncertainty associated with wind speed and solar irradiation. The scenario-tree technique was used to model the uncertainties associated with solar radiation, wind speed and load demand [28]. This approach is found to be effective while handling uncertainties but requires heavy calculation. An optimal generation scheduling strategy with total contributions of wind farms, solar parks and thermal plants for economic benefit and environmental impact is presented in reference [29]. Here uncertainty associated with wind, solar power and few coal units is described by fuzzy numbers.

The techno-economic analysis under distinct scenarios has been investigated with different combinations of renewable energy resources. Results show that integrating multiple RER and its appropriate scheduling helps minimize cost and emission [30]. A detailed review of various computation methods for planning a hybrid renewable system is presented in reference [31]. Inspired by distinct work carried out by various researchers, in this paper, a novel optimization method incepted in 2020, Equilibrium Optimizer (EO) [32], inspired by dynamic and equilibrium states of physics, is used to solve the complex multi-objective problem with and without integration of RES. The schematic diagram of the proposed model is shown in Figure 1.

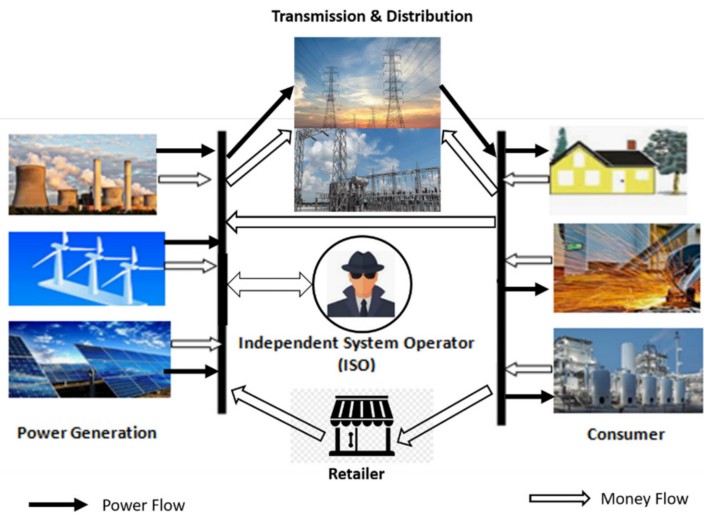

**Figure 1.** The grid with RES under deregulated environment.

The main contribution in the paper is as follows: (i) Equilibrium Optimizer (EO) is implemented to solve the multi-objective problem in a dynamic environment, (ii) impact of RER integration was analyzed in terms of reduction in operational cost and emission (iii) fuzzy-min ranking method is used to aggregate two conflicting objectives and finally, (iv) profit achieved by the reduction in operational cost due to RER integration were analyzed.

The remaining paper organization is as follows: problem formulation is given in Section 2, the optimization method is described in Section 3, results and discussion are presented in Section 4 and the conclusions are summarized in Section 5.

## 2. Formulation of the Profit-Based Multi-Objective Scheduling Problem

### 2.1. Objective Function I: Profit Maximization

The difference between revenue (*RV*) collected from the sale of electricity and the total cost of electrical power generation (*TC*) is taken as profit [33]. The objective is to maximize the profit defined as (1):

$$Max(Profit) = Max\ (RV - TC) \tag{1}$$

where,

$$RV = \sum_{t=1}^{24} \left( \sum_{j=1}^{NT} Gth_{jt} + \sum_{k=1}^{NS} Gs_{kt} + \sum_{l=1}^{NW} Gw_{lt} \right).SP_t \tag{2}$$

$$TC = \sum_{t=1}^{24} \left( \sum_{j=1}^{NT} C(Gth)_{jt} + \sum_{k=1}^{NS} C(Gs)_{kt} + \sum_{l=1}^{NW} C(Gw)_{lt} \right) \tag{3}$$

The first part of (3) is the generation cost of thermal units defined in (4) as:

$$C(Gth)_{jt} = \left( a_j \times Gth_{jt}^2 + b_j \times Gth_{jt} + c_j \right) \tag{4}$$

where $Gth_{jt}$ is generated power by $j^{th}$ thermal unit at $t^{th}$ time interval, $a_j$, $b_j$, $c_j$ are fuel coefficients of $j^{th}$ thermal unit, $SP_t$ is the selling price at $t^{th}$ time interval, *NT, NS* and *NW*, are the number of thermal, solar and wind units present in the hybrid system [34]. Total 24-time steps representing 24 h in a day are considered.

The second part represents the generation cost of the solar PV system. Solar power output depends on incident solar radiation ($R_t$) and the difference of ambient ($\theta_{amb}$) and reference temperature ($\theta_{ref}$). Therefore, it increases the uncertainty in computing the availability of the solar power output. Thus, underestimation and overestimation cost is added here to balance the variation due to uncertainty.

The solar power output of $k^{th}$ plant at time $t$, ($Gs_{kt}$) is given as [34]:

$$Gs_{kt} = Pr\left\{ \left( 1 + \left( \theta_{amb} - \theta_{ref} \right)\tilde{\xi} \right) \times (R_t/1000) \right\} \tag{5}$$

Total solar cost function at any time $t$ is calculated as [34]:

$$\sum_{k=1}^{NS} C(Gs_{kt}) = \sum_{k=1}^{NS}(DC_s \times Gs_{kt}) + \sum_{k=1}^{NS} k_p(Gsav_{kt} - Gs_{kt}) + \sum_{k=1}^{NS} k_r(Gs_{kt} - Gsav_{kt}) \quad (6)$$

where,

$$k_p \times (Gsav_{kt} - Gs_{kt}) = k_p \times \int_{Gs_{kt}}^{P_r} (s - Gs_{kt})f_s(s)ds \quad (7)$$

$$k_r \times (Gs_{kt} - Gsav_{kt}) = k_r \times \int_{0}^{Gs_{kt}} (Gs_{kt} - s)f_s(s)ds \quad (8)$$

$k_p$, $k_r$ represents the penalty cost factor for overestimation and underestimation, $Gs_{kt}$, $Gsav_{kt}$ are the generated and available power for $k^{th}$ solar plant at $t^{th}$ time, respectively, and $f_S(s)$ *is* probability density function ($pdf$) of solar power.

The third part of (3) represents the cost due to wind power integration. Wind power depends on wind velocity, and at any given location, it is observed to follow the Weibull probability distribution [35], as shown in Figure 2. The random wind speed variable can be computed using the Weibull $pdf$ as:

$$f_v(v) = \left(\frac{k}{c}\right)\left(\frac{v}{c}\right)^{k-1} e^{-(vc)^k} \quad 0 < v < \infty \quad (9)$$

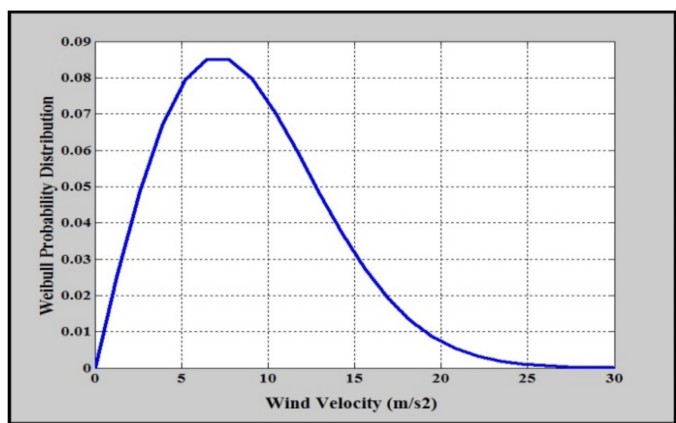

**Figure 2.** The Weibull probability distribution for wind velocity.

Wind power ($Gw$) at different velocities is calculated as:

$$Gw = \begin{cases} 0 & for \quad 0 \le v < v_{ci} \\ Wr\left(\frac{v - v_{ci}}{v_r - v_{ci}}\right) & for \quad v_{ci} \le v < v_r \\ Wr & for \quad v_r \le v < v_{co} \\ 0 & for \quad v > v_{co} \end{cases} \quad (10)$$

where, $v$, $v_{ci}$, $v_{co}$ *and* $v_r$, are the wind velocity at any instant, cut-in speed, cut-out speed and rated wind turbine speed in m/s, respectively. $Wr$ is the rated power of wind turbine in MW [36]. The wind speed cost has three terms; the first term is a fixed cost term while the second and third terms arise due to the uncertainty involved with wind power generation.

After including the over and under estimation costs, the total cost of $l^{th}$ wind plant at time $t$ can be mathematically expressed as:

$$\sum_{l=1}^{NW} C(Gw_{lt}) = \sum_{l=1}^{NW}(DC_w \times Gw_{lt}) + \sum_{l=1}^{NW} k_p(Gwav_{lt} - Gw_{lt}) + \sum_{l=1}^{NW} k_r(Gw_{lt} - Gwav_{lt}) \quad (11)$$

The second and third terms in (11) represent the penalty cost due to underestimation and reserve cost due to overestimation and are expressed as (12) and (13), respectively.

$$k_p \times (Gwav_{lt} - Gw_{lt}) = k_p \times \int_{Gw_{lt}}^{W_r} (w - Gw_{lt}) f_W(w) \qquad (12)$$

$$k_r \times (Gw_{lt} - Gwav_{lt}) = k_r \times \int_{0}^{Gw_{lt}} (Gw_{lt} - w) f_W(w) dw \qquad (13)$$

### 2.2. Objective Function II: Emission Minimization

The total emission (*TE*) emitted from the thermal power plant [37] can be symbolically represented as:

$$\text{Min}(TE) = Min\left\{ \sum_{j=1}^{NT} E(Gth) \right\} \qquad (14)$$

where,

$$E(Gth)_j = \left( \alpha_j \times Gth_j^2 + \beta_j \times Gth_j + \gamma_j \right) \qquad (15)$$

Here $\alpha_j, \beta_j, \gamma_j$ are the emission coefficients of $j^{th}$ thermal unit.

### 2.3. Objective Function III: Simultaneous Optimization of Profit and Emission

The multi-objective problem of optimization of profit and emission can be formulated as:

$$\text{Min}\left\{ \left( \frac{1}{Profit} \right),\ TE \right\} = \frac{u}{\sum_{t=1}^{24} \left( \sum_{j=1}^{NT} Gth_{jt} + \sum_{k=1}^{NS} Gs_{kt} + \sum_{l=1}^{NW} Gw_{lt} \right). SP_t - \sum_{t=1}^{24} \left( \sum_{j=1}^{NT} C(Gth)_{jt} + \sum_{k=1}^{NS} C(Gs)_{kt} + \sum_{l=1}^{NW} C(Gw)_{lt} \right)} + (1-u).ppf.\ \sum_{j=1}^{NT} E(Gth) \qquad (16)$$

where $ppf$ is the price penalty factor which is assumed to be 1 for simplicity and u is the weight factor. Maximization of profit is an objective function I and minimization of total emission is objective function II. Profit and cost of generation are two cornerstones of the electrical market. A decrease in the cost of generation will increase profit. This reciprocal relation between cost and profit is modelled as objective function III or simultaneous optimization of profit and emission.

It is subjected to the following operational constraints:

#### 2.3.1. Real Power Balance: Power Demand at Any Instant (t) Must Be Equal to the Sum of the Power Output of Associated Generation Units

$$PD(t) = \sum_{j=1}^{NT} Gth_j + \sum_{k=1}^{2NS} Gs_k + \sum_{l=1}^{NW} Gw_l \qquad (17)$$

#### 2.3.2. Generation Limit: The Power Produced by Each Thermal, Wind and the Solar Unit Must Always Be between Their Respective Specified Bounds, as Given by

$$Gth_j^{min} \leq Gth_j \leq Gth_j^{max} \qquad (18)$$

$$Gs_k^{min} \leq Gs_k \leq Gs_k^{max} \qquad (19)$$

$$Gw_l^{min} \leq Gw_l \leq Gw_l^{max} \qquad (20)$$

#### 2.3.3. Ramp-Rate Limit Constraints: The Thermal Units Have Limited Ramping (Up as Well as Down) Capacity, and Therefore the Output of a Unit between two Consecutive Time-Intervals Must Obey the Inequality Constraint Given by

$$Gth_{jt} - Gth_{j(t-1)} \leq UR_j \qquad (21)$$

$$Gth_{j(t-1)} - Gth_{jt} \leq DR_j \qquad (22)$$

Here, $UR_j,\ DR_j$ are the up and down rates of the $j^{th}$ thermal unit.

Combining the ramp rate limits of generating units given by (21) and (22) with the thermal generation limits provided by (18), the modified binding constraints can be written as:

$$Max\left(Gth_j^{min}, Gth_{j(t-1)} - DR_j\right) \leq Gth_{jt} \leq Min\left(Gth_j^{min}, Gth_{j(t-1)} + UR_j\right) \tag{23}$$

### 2.4. Ranking Approach

The fuzzy-min ranking method is used to aggregate two conflicting objectives: profit and emission [38]. Linear membership function, $\mu_{i,r}$ ($r^{th}$ is the objective function for the $i^{th}$ solution) is described for each objective function in (24) as [22].

$$\mu_{i,r} = \begin{cases} 1 & if \quad F_{i,r} \leq F_r^{min} \\ \frac{F_r^{max} - F_{i,r}}{F_r^{max} - F_r^{min}} & if \quad F_r^{min} \leq F_{i,r} \leq F_r^{max} \\ 0 & if \quad F_{i,r} \geq F_r^{max} \end{cases} \tag{24}$$

For $i^{th}$ solution with $n$ number of objectives, the rank is computed as:

$$fuzzy\_rank_i = min(\mu_{i,r}) \quad for \quad r = 1, 2 \ldots .n \tag{25}$$

The solution with the maximum value of $fuzzy\_rank$ for $\forall$r is considered as the best compromise solution.

## 3. Equilibrium Optimization

Equilibrium Optimization is a physics-based algorithm that follows the concept of dynamic mass balance is given control space. The first-order differential equation, which relates the mass-generated in a dynamic system with mass entering and mass leaving the system, can be expressed as:

$$V\frac{d\mathbb{C}}{dt} = Q\mathbb{C}_{eq} - Q\mathbb{C} + \mathcal{G} \tag{26}$$

where, $\mathbb{C}$ is the concentration in volume ($V$), $V\frac{d\mathbb{C}}{dt}$ is the rate of change in mass, $Q$ is the flow rate, $\mathbb{C}_{eq}$ is the concentration at equilibrium state and $\mathcal{G}$ represents the rate of generation of mass. The equilibrium state is supposed to be achieved whenever $V\frac{d\mathbb{C}}{dt}$ reaches zero. The derivative $\frac{d\mathbb{C}}{dt}$, can be solved as a function of $\left(\frac{Q}{V}\right)$. The ratio $\frac{Q}{V} = \mu$ is called the turnover rate.

Equation (26) can be rearranged and written as:

$$\frac{d\mathbb{C}}{\mu\mathbb{C}_{eq} - \mu\mathbb{C} + \frac{\mathcal{G}}{V}} = dt \tag{27}$$

and,

$$\int_{\mathbb{C}_o}^{\mathbb{C}} \left(\frac{d\mathbb{C}}{\mu\mathbb{C}_{eq} - \mu\mathbb{C} + \frac{\mathcal{G}}{V}}\right) = \int_{t_o}^{t} dt \tag{28}$$

The final concentration update equation after rearranging and integrating becomes [32]:

$$\mathbb{C} = \mathbb{C}_{eq} + \left(\mathbb{C}_o - \mathbb{C}_{eq}\right)\mathcal{F} + \frac{\mathcal{G}}{\mu V}(1 - \mathcal{F}) \tag{29}$$

where,

$$\mathcal{F} = exp\{-\mu(t - t_o)\} \tag{30}$$

Equation (29) provides the search mechanism for finding an optimal solution during the optimization process of EO. Here $\mathbb{C}_{eq,}$ is a solution that is selected randomly from a pool consisting of 3 to 5 best solutions collected after solving the problem for different conditions. The second term $\left(\mathbb{C}_o - \mathbb{C}_{eq}\right)$ is the difference in the position of a solution and the randomly selected equilibrium state. This term provides direct search and persuades particles to conduct a global search and explore the solution space extensively and effectively. The

third term $\left\{\frac{G}{\mu V}(1 - \mathcal{F})\right\}$ is the term associated with the generation rate and turnover rate. This term improves/updates the solution through exploitation; hence the steps are small, resulting in small changes to fine-tune the solution; however, sometimes, it allows exploration. EO follows five steps during the optimization process described as follows:

### 3.1. Initialization

The initialization procedure in EO is similar to other population-based metaheuristics. The initial population is created by randomly generating concentrations within the minimum and maximum limits for each dimension of the vector. The $i^{th}$ population vector can be constructed as follows:

$$\mathbb{C}_i^{initial} = \mathbb{C}_{min} + rand(\mathbb{C}_{max} - \mathbb{C}_{min}) \tag{31}$$

Here $\mathbb{C}_{max}$ and $\mathbb{C}_{min}$, represents vectors representing maximum and minimum concentrations of the different dimensions of the solution vector. The generated particles (solutions) are evaluated, and their fitness value is determined. Then the equilibrium pool is set up by using 3 to 5 best solutions.

### 3.2. Equilibrium Pool and Candidates

The equilibrium state is the global best solution of the problem, which is obtained after convergence. The equilibrium pool is created by storing the best solutions of runs conducted under different conditions. The arithmetic mean of these best solutions is also stored in the pool as shown:

$$\vec{\mathbb{C}}_{eq,pool} = \left\{\vec{\mathbb{C}}_{eq,1}, \vec{\mathbb{C}}_{eq,2}, \vec{\mathbb{C}}_{eq,3}, \vec{\mathbb{C}}_{eq,4}, \vec{\mathbb{C}}_{eq,ave}\right\} \tag{32}$$

For updating the position of a particle using (29), one of these best solutions from (32) is randomly selected. The probability of selection is uniform for all equilibrium concentrations of the pool. Suppose there are 5 candidate solutions as shown above. In that case, the new solution will be generated by exploration if any of the first four equilibrium states/solutions in the pool are selected for the position update mechanism. On the other hand, if the fifth candidate is chosen for position update, then exploitation is carried out to generate a new solution/state.

### 3.3. Exponential Term

The third term in the concentration update Equation (29) is the exponential term ($\mathcal{F}$). This term is designed to provide an effective balance between exploration and exploitation in the EO algorithm.

$$\vec{\mathcal{F}} = exp\left(-\vec{\mu}(t - t_0)\right) \tag{33}$$

The turnover rate ($\mu$) is a random number ranging from 0 to 1. Time is represented by $t$, which decreases with iteration, as expressed below.

$$t = \left(1 - \frac{Iter}{Max\_iter}\right)^{a_2 \times \frac{Iter}{Max\_iter}} \tag{34}$$

$$\vec{t_0} = \frac{1}{\vec{\mu}} ln\left\{-a_1 sign\left(\vec{r} - 0.5\left[1 - e^{-\vec{\mu}t}\right]\right) + t\right\} \tag{35}$$

Substituting of (35) in (33) gives the final expression for the exponential term ($\mathcal{F}$) presented in Equation (36). The plot of $\mathcal{F}$ for four different combinations of $a_1$ and $a_2$ is shown in Figure 3. The exponential term ($\mathcal{F}$) variation with iteration can be seen to decrease (in both directions) and finally converge to zero for all four combination cases. The nature of variation indicates the effectiveness of this term in conducting exploration and exploitation.

$$\vec{\mathcal{F}} = a_1 sign\left(\vec{r} - 0.5\right)\left[e^{-\vec{\mu}t} - 1\right] \tag{36}$$

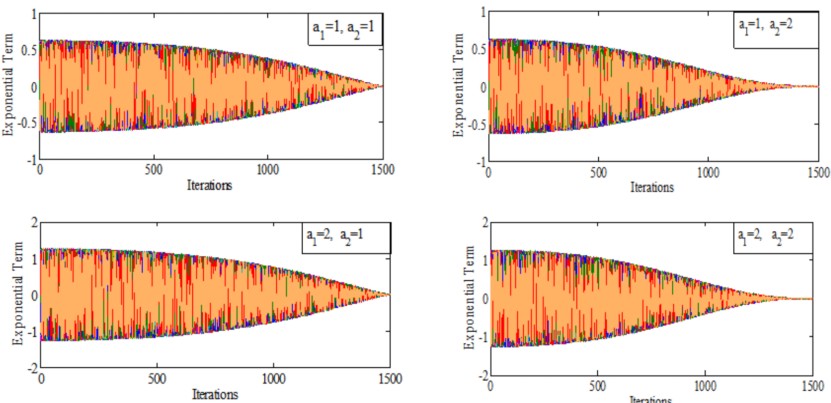

**Figure 3.** Variation of the exponential term for different combinations of $a_1$ and $a_2$.

In EO exploration and exploitation are controlled by the constants $a_1$ and $a_2$, respectively. Other than these two, the term $sign\left(\vec{r} - 0.5\right)$ affects the direction of exploration and exploitation during the search.

### 3.4. Generation Rate

This term constitutes the third term of the concentration update equation given by (29). The generation rate ensures the convergence of the algorithm to the optimal global solution $(\vec{\mathcal{G}})$ which facilitates smooth convergence by tuning the solutions using small updates. By modeling the generation rate using an exponential decay term of the first order and assuming the decay constant to be equal to the turnover rate [32], the generation rate can be expressed to be decreasing from an initial value $(\mathcal{G}_0)$ as:

$$\vec{\mathcal{G}} = \vec{\mathcal{G}_0}e^{-\vec{\mu}(t-t_0)} = \vec{\mathcal{G}_0}\vec{\mathcal{F}} \tag{37}$$

$$\vec{\mathcal{G}_0} = \vec{\lambda}\left(\vec{\mathbb{C}_{eq}} - \vec{\mu}\vec{\mathbb{C}}\right) \tag{38}$$

The generation rate control parameter $(\lambda)$ decides what role will be played by the generation rate term in updating the particle position in (29). This parameter is designed to controls the exploitation and exploration of the particle as follows:

$$\vec{\lambda} = \begin{cases} 0.5r_1 & r_2 > \rho \\ 0 & r_1 < \rho \end{cases} \tag{39}$$

The probability of using the generation rate term by the particle while updating its concentration using (29) is also determined by the generation probability expressed by $(\rho)$.

Here $r_1$ and $r_2$ are uniformly distributed random numbers in $[0, 1]$. If the first condition in (39) is true, then the generation rate parameter will be small, and the update step size will be small, causing exploitation. But if the second condition is true, then the particle is updated without any contribution from the generation rate term as $(\lambda)$ and $(\mathcal{G})$ both become zero. Experiments have shown that when $(\rho)$ is set at 0.5, the search undergoes balanced exploitation and exploration. As the generation probability $\rho$ is increased beyond 0.5, exploration increases, and as $\rho$ is decreased below 0.5, exploitation is observed to increase.

### 3.5. Particle's Memory Saving

In random operator-based optimization algorithms, some kind of memory mechanism must be used to avoid losing a better solution during the process. EO also has a procedure somewhat similar to the *pbest* in PSO, where the best position and corresponding fitness of each particle are stored and updated whenever there is an improvement in subsequent iterations. The flow chart of EO is shown in Figure 4.

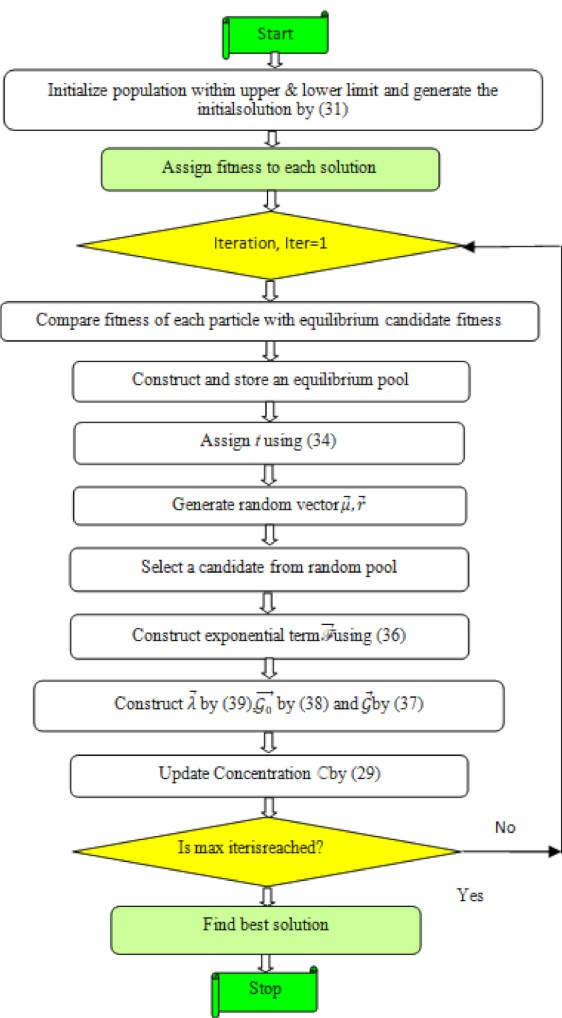

**Figure 4.** Flow chart of Equilibrium Optimization.

*3.6. Steps for Implementation of EO for Profit Based Generation Scheduling*

**Step 1:** Define population size, the maximum number of iterations and the number of runs for the algorithm.

**Step 2:** Initialize the population within lower and upper concentration limits following (31) and check the constraints (18)–(20) and (23).

**Step 3:** Select the equilibrium candidates from the initial population assign $a_1$, $a_2$, $\rho$ and evaluate the values of each equilibrium candidate.

**Step 4:** Compare the fitness of equilibrium candidates with that of each particle in the updated population. Replace the equilibrium candidate and corresponding fitness value with that of the particle from the population, if fitness is better.

**Step 5:** Continue step 4 in loops equal to the number of particles in the population. At the end of the loop, one will acquire the Equilibrium pool $\overrightarrow{\mathbb{C}}_{eq,pool}$ as in (32)

**Step 6:** Accomplish memory saving

**Step 7:** Construct '$t$' as per (34)

**Step 8:** Run a loop equal to the number of particles in the population and randomly choose one candidate from the equilibrium pool (vector).

**Step 9:** Construct $\overrightarrow{\mu}$ ,$r_1$,$\overrightarrow{\mathcal{F}}$, $\overrightarrow{\mathcal{G}}$ ,$\overrightarrow{\mathcal{G}_0}$, $\overrightarrow{\lambda}$ using (36)–(39)

**Step 10:** Update concentrations using (29) till the number of iterations is less than the maximum number of iterations. After the loop is finished, the final concentrations are the power output for the thermal generators for the given time period of 24 h.

**Step 11:** Apply a Fuzzy selection mechanism to find out the best compromise solution

**Step 12:** Store the best compromise solution.

## 4. Results and Discussion

The performance of the EO algorithm is tested on standard test cases under dynamic constraints [38–40]. Impact on operational cost and emission due to RER integration are also investigated here. The objective function is written in MATLAB R2013a environment and executed on Intel core i7 processor with 2 GB RAM and 3.40 GHz speed.

### 4.1. Description of Test Cases

**Test Case 1** has six thermal power generating unit system. Its selling cost, minimum and maximum power limits and cost/emission coefficients are listed in Table A1 [40], along with power demand and hourly selling prices on a particular day.

**Test Case 2** is a modified test case created by adding two solar PV units to test case 1. The data of solar plants are listed in Table A2. Its data related to radiation and corresponding temperature are shown in Figure 5.

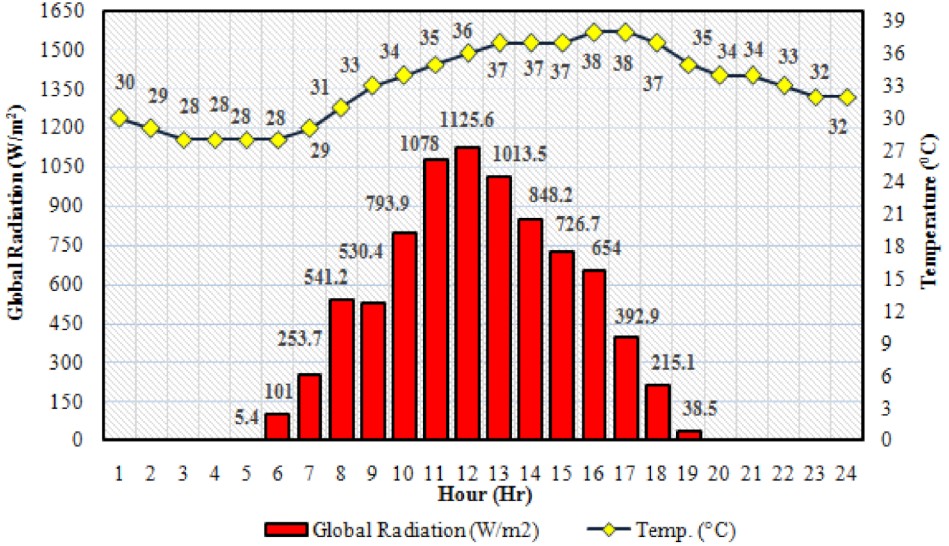

**Figure 5.** Solar PV data of Temperature (°C) and Radiation (W/m$^2$).

**Test Case 3** is also a modified test case obtained by the addition of two wind generators to test case 1. The data for wind generators are listed in Table A2.

**Test Case 4** is a hybrid thermal-wind-solar PV system that integrates two wind generators and two solar PV systems with thermal units of Test case 3, Test case 2 and Test case 1, respectively.

### 4.2. Effect of Number of Particles

To analyze optimal particle size ($NP$), experiments are conducted on Test Case 1 with different values of $NP$. Its effect on optimal generation cost is plotted in Figure 6. Here, it is observed that with an $NP$ of 200, the mean operation cost is the lowest. By increasing $NP$ beyond it, no significant change was observed; however, computational time increases. Hence, a particle size of 200 is considered for further analysis of the problem. The performance of EO is also validated by the comparison of results obtained by simulation of two well-established algorithms: particle swarm optimization (PSO) and artificial bee colony (ABC) algorithm, keeping the same population size. The statistical results in terms of operational cost are tabulated in Table 1 over 30 repeated runs. The cost convergence curve of the three algorithms is compared in Figure 7. The above two results show the superiority of EO over the other two in terms of better search capability and fast convergence.

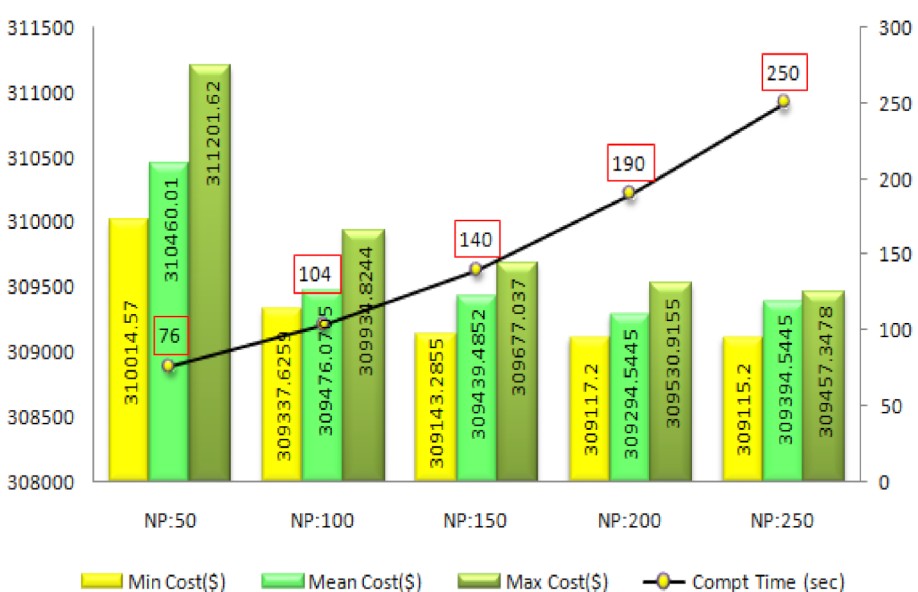

**Figure 6.** Effect of Number of particles (NP).

**Table 1.** Comparison of EO with ABC and PSO for Test Case 1.

| Method | Parameters | Min Cost ($) | Mean Cost ($) | Max Cost ($) | SD | CPU Time/Iter. (s) |
|---|---|---|---|---|---|---|
| PSO | $c_1 = c_2 = 2.1\ w_{min} = 0.4$ $w_{max} = 0.9$ | 309,125.58 | 309,133.97 | 309,170.86 | 5.3818 | 0.0215 |
| ABC | Limit=100 | 309,126.34 | 309,154.07 | 309,164.77 | 10.05 | 0.0203 |
| EO | $a_1$=2, $a_2$=1, ρ=0.5 | 309,117.20 | 309,125.54 | 309,139.91 | 0.9103 | 0.0188 |

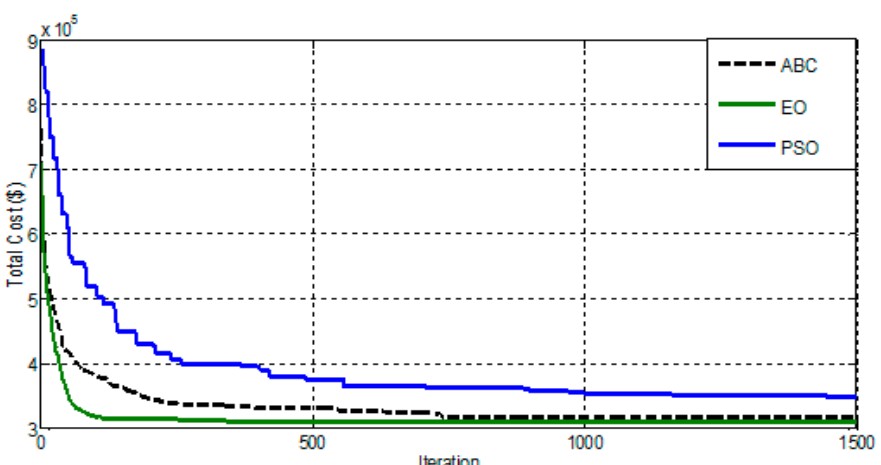

**Figure 7.** Cost Convergence curve of ABC, PSO and EO for test case 1.

### 4.3. Effect of Control Parameters on the Performance of EO

EO has three control parameters:

- constants $a_1$ and $a_2$ which control the exploration and exploitation, and
- generation probability (ρ) decides whether exploration or exploitation of search space will occur.

To analyze the impact of the above three control parameters, various tests are conducted on Test Case 4 with the variation of parameters in the prescribed range, and obtained results are tabulated in Tables 2 and 3. In Table 2, the value of ρ is kept at 0.5, and results are

computed for different combinations of constants $a_1$ and $a_2$. Here, it is observed that best results are obtained when $a_1 = 2$ and $a_2 = 1$. Further $\rho$ is varied from 0.1 to 0.9, keeping $a_1 = 2$ and $a_2 = 1$ fixed and obtained simulation results are summarised in Table 3. It is evident that the best result, with the highest profit and lowest emission content, is obtained with $\rho = 0.5$ where the exploration and exploitation have an equal chance of occurrence. Therefore, these combinations of control parameters are considered for further analysis.

**Table 2.** Best Compromise Solution with the variation of parameters $a_1$, $a_2$ for Test Case 4 ($\rho = 0.5$).

| Parameters | | Total Cost ($) | Profit ($) | Emission (Kg) |
|---|---|---|---|---|
| $a_1$ | $a_2$ | | | |
| 1 | 1 | 301,550.07 | 337,807.18 | 24,412.00 |
| 1 | 2 | 309,167.46 | 330,189.73 | 28,484.83 |
| **2** | **1** | **297,031.57** | **342,325.68** | **24,763.26** |
| 2 | 2 | 297,620.53 | 341,736.72 | 26,759.37 |

**Table 3.** Best Compromise Solution with variation of generation probability '$\rho$' for Test Case 4 ($a_1 = 2$ and $a_2 = 1$).

| | $\rho$ | Total Cost ($) | Profit ($) | Emission (Kg) |
|---|---|---|---|---|
| | 0.1 | 298,680.22 | 340,677.03 | 28,639.24 |
| | 0.2 | 298,215.49 | 341,141.76 | 27,678.89 |
| | 0.3 | 298,261.84 | 341,095.41 | 27,217.14 |
| | 0.4 | 298,310.80 | 341,046.45 | 28,180.66 |
| $a_1 = 2$, $a_2 = 1$ | 0.5 | 297,031.57 | 342,325.68 | 24,763.26 |
| | 0.6 | 297,845.79 | 341,511.46 | 27,399.93 |
| | 0.7 | 297,553.80 | 341,803.45 | 26,343.26 |
| | 0.8 | 297,624.74 | 341,732.51 | 26,728.33 |
| | 0.9 | 298,367.13 | 340,990.12 | 26,233.64 |

### 4.4. Effect of RER Integration on Profit Maximization

The simulation results for different test cases under the scenario of cost minimization/profit maximization given by (1) are shown in Figure 8. Comparing test case 1 and test case 2 shows a reduction in power generation cost by 3883.3 $ (≈1.26%), and hence the profit is increased by 3883.3 $ (≈1.18%). While comparison of test case 1 and test case 3, it is observed that the reduction in power generation cost is found to be 13,461.09 $ (≈4.35%), and hence the profit is increased by 13,461 $ (≈4.08%). Similarly, a comparison of test case 1 and the hybrid test case 4, shows that the power generation cost is reduced by 14,285.21 $ (≈4.62%) and the profit increased by 14,285.18 $ (≈4.32%). Hence, it is clear that the higher the integration of RER, the higher the profit even after the inclusion of cost due to uncertainty.

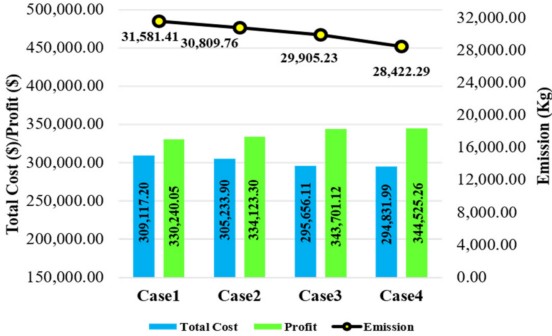

**Figure 8.** Comparison of cost, profit and emission for profit maximization.

The optimum generation schedule for test cases 1 and 4 are shown in Figures 9 and 10, where operational constraints are fully satisfied. Profit and emission for these two cases are compared and listed in Table 4.

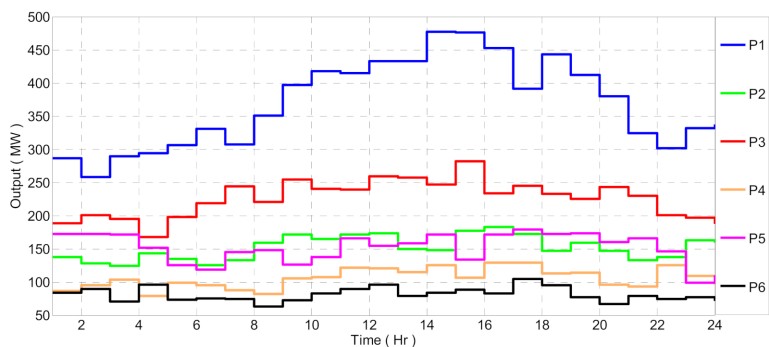

**Figure 9.** Optimal generation schedule under the scenario of Cost minimization (Test Case 1).

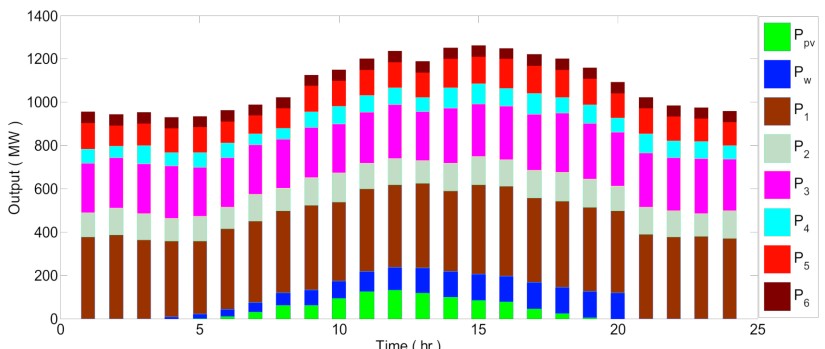

**Figure 10.** Optimal generation schedule under the scenario of Cost minimization (Test Case 4).

**Table 4.** Cost, Selling price, Profit and Emission under the scenario of profit maximization.

|  | ThC ($) | WC ($) | PVC ($) | TC ($) | SP ($) | Profit ($) | Emission (Kg) |
|---|---|---|---|---|---|---|---|
| Test Case 1 | 309,117.20 | – | – | 309,117.20 | 639,357.25 | 330,240.05 | 31,581.41 |
| Test Case 4 | 276,906.41 | 6290.20 | 11,635.38 | 294,831.99 | 639,357.25 | 344,525.26 | 28,422.29 |

### 4.5. Effect of RER Integration on Emission Minimization

For minimization of emission (14), all the test cases under consideration are carefully analyzed to find the impact of the integration of (i) solar, (ii) wind and (iii) both solar and wind resources. The objective is to determine the optimal schedule for all four test cases, which will produce minimum emission content. The results for each test case are presented in Figure 11.

Comparing test case 1 and case 2, the reduction in emission content is 2255.94 kg ($\approx$8.92%) due to the solar share of 975.71 MW (4%). While Comparing test case 1 and test case 3, the reduction in emission content is 3216.74 kg ($\approx$12.72%) due to 1463.08 MW (5%) of wind share. Similarly, while comparing test case 1 and test case 4, emission reduction is 5524.63 kg ($\approx$21.84%).

Comparing all the cases, it is observed that there is a significant reduction in pollution by integration of RER.

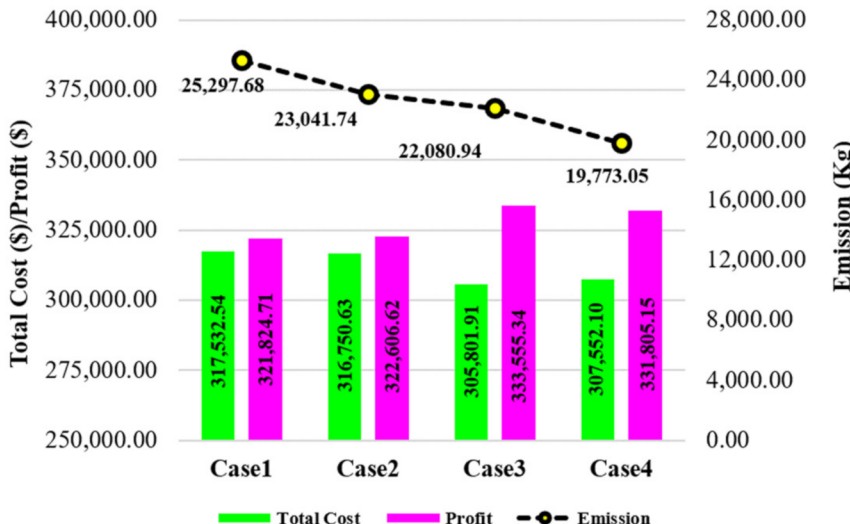

**Figure 11.** Comparison of cost, profit and emission under the scenario of emission minimization.

### 4.6. Effect of RER Integration for the Multiobjective Case

For the simultaneous optimization of profit and emission both, formulated in (16) with the help of the fuzzy min approach (24), the best compromise solution is obtained. Pareto front for all test cases obtained by EO is plotted and compared in Figure 12 and the top 10 optimal solutions and their fuzzy min rank are tabulated in Table 5. The best-compromised solution with the highest fuzzy rank is indicated for each test case.

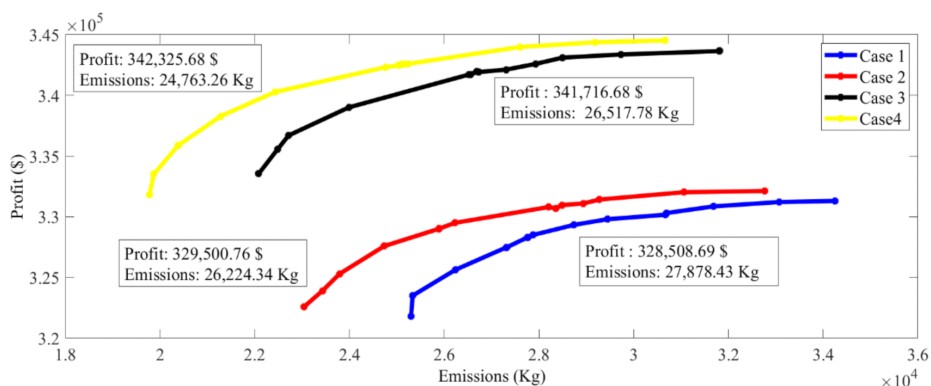

**Figure 12.** Pareto-front of all the non-dominated solutions obtained for different cases.

**Table 5.** Top 10 optimal solutions and their fuzzy min rank.

| Test Case 1 | | | | | Test Case 2 | | | | |
|---|---|---|---|---|---|---|---|---|---|
| **Profit** | **Emission** | $\mu_1$ | $\mu_2$ | **min ($\mu$)** | **Profit** | **Emission** | $\mu_1$ | $\mu_2$ | **min ($\mu$)** |
| 328,508.692 | 27,878.429 | 0.705 | 0.712 | 0.705 | 329,500.761 | 26,224.344 | 0.725 | 0.672 | 0.672 |
| 328,312.919 | 27,759.219 | 0.684 | 0.725 | 0.684 | 329,502.218 | 26,226.754 | 0.725 | 0.672 | 0.672 |
| 328,312.663 | 27,759.160 | 0.684 | 0.725 | 0.684 | 329,503.106 | 26,227.526 | 0.725 | 0.672 | 0.672 |
| 328,311.440 | 27,758.630 | 0.684 | 0.725 | 0.684 | 327,594.083 | 24,734.770 | 0.524 | 0.825 | 0.524 |
| 329,335.245 | 28,738.401 | 0.792 | 0.616 | 0.616 | 327,593.259 | 24,733.423 | 0.524 | 0.825 | 0.524 |
| 329,336.053 | 28,739.487 | 0.792 | 0.616 | 0.616 | 327,591.770 | 24,732.427 | 0.524 | 0.826 | 0.524 |
| 329,336.009 | 28,739.710 | 0.792 | 0.616 | 0.616 | 330,805.950 | 28,204.049 | 0.862 | 0.468 | 0.468 |
| 327,479.749 | 27,313.518 | 0.596 | 0.775 | 0.596 | 330,805.924 | 28,204.710 | 0.862 | 0.468 | 0.468 |
| 327,478.885 | 27,312.756 | 0.596 | 0.775 | 0.596 | 330,805.931 | 28,205.116 | 0.862 | 0.468 | 0.468 |
| 327,478.002 | 27,312.382 | 0.596 | 0.775 | 0.596 | 330,689.142 | 28,348.341 | 0.849 | 0.454 | 0.454 |
| Test Case 3 | | | | | Test Case 4 | | | | |
| **Profit** | **Emission** | $\mu_1$ | $\mu_2$ | **min ($\mu$)** | **Profit** | **Emission** | $\mu_1$ | $\mu_2$ | **min ($\mu$)** |
| 341,716.684 | 26,517.776 | 0.760 | 0.567 | 0.567 | 342,325.682 | 24,763.258 | 0.827 | 0.542 | 0.542 |
| 341,716.956 | 26,518.660 | 0.760 | 0.567 | 0.567 | 342,325.842 | 24,763.330 | 0.827 | 0.542 | 0.542 |
| 341,717.139 | 26,518.991 | 0.760 | 0.567 | 0.567 | 342,477.714 | 25,040.173 | 0.839 | 0.516 | 0.516 |
| 341,708.322 | 26,549.079 | 0.759 | 0.564 | 0.564 | 342,478.170 | 25,040.791 | 0.839 | 0.516 | 0.516 |
| 341,708.646 | 26,549.144 | 0.759 | 0.564 | 0.564 | 342,532.703 | 25,090.462 | 0.843 | 0.512 | 0.512 |
| 341,708.329 | 26,549.322 | 0.759 | 0.564 | 0.564 | 342,533.058 | 25,090.864 | 0.843 | 0.512 | 0.512 |
| 341,960.680 | 26,683.972 | 0.791 | 0.550 | 0.550 | 338,289.653 | 21,279.033 | 0.510 | 0.862 | 0.510 |
| 341,960.976 | 26,684.062 | 0.791 | 0.549 | 0.549 | 338,288.413 | 21,278.272 | 0.510 | 0.862 | 0.510 |
| 341,961.230 | 26,684.330 | 0.791 | 0.549 | 0.549 | 342,585.154 | 25,130.773 | 0.847 | 0.508 | 0.508 |
| 341,924.268 | 26,723.629 | 0.786 | 0.545 | 0.545 | 342,584.919 | 25,130.786 | 0.847 | 0.508 | 0.508 |

The optimal power generation schedule for test case 1 and hybrid thermal-wind-solar PV system, i.e., test case 4 with cost, selling price, profit and corresponding emission are listed in Tables 6 and 7. In addition, a comparison of cost, profit and emission for all four cases is presented in Figure 13. Here it is seen that the cost is reduced by 992.07 $ ($\approx$0.32%) after integration of two solar units in test case 2. The cost was reduced by 13,207.97$ ($\approx$4.25%) in test case 3 when two wind power units were added and by 13,816.99 $ ($\approx$4.44%) for test case 4 when two solar and wind units, respectively, were integrated with the existing thermal system. The profit is found to increase by 992.07 $ ($\approx$0.30%), 13,208$ ($\approx$4.02%), 13,816.66 $ ($\approx$4.21%), respectively, for test cases 2, 3 and 4.

The emission content is observed to reduce by 1654.09 kg ($\approx$5.93%) in case 2, by 1360.65 kg ($\approx$4.88%) in case 3, and by 3115.17 kg ($\approx$11.17%) in case 4 with respect to test case 1 where only thermal units are present.

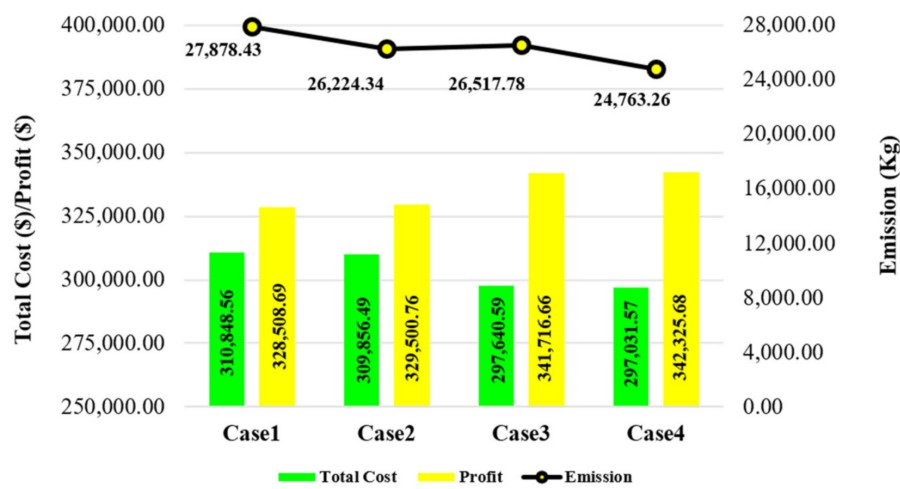

**Figure 13.** Comparison of cost, profit and emission under the scenario of emission minimization.

**Table 6.** Optimal generation schedule, cost, selling price and profit for Test Case 1 (Best Compromise Solution).

| Hour | P1 (MW) | P2 (MW) | P3 (MW) | P4 (MW) | P5 (MW) | P6 (MW) | TC ($) | SP ($) | Profit ($) | Emission (Kg) |
|------|---------|---------|---------|---------|---------|---------|--------|--------|------------|---------------|
| 1 | 267.18 | 116.94 | 189.30 | 137.60 | 125.15 | 118.82 | 11,431.4097 | 21,630.75 | 10,199.34 | 884.64 |
| 2 | 290.12 | 120.26 | 180.63 | 108.07 | 135.22 | 107.69 | 11,208.0562 | 20,724.00 | 9515.94 | 900.50 |
| 3 | 273.65 | 147.61 | 201.12 | 89.01 | 122.96 | 118.66 | 11,357.6823 | 21,537.80 | 10,180.12 | 920.34 |
| 4 | 258.12 | 147.41 | 159.08 | 113.96 | 146.97 | 104.46 | 11,145.9213 | 20,553.00 | 9407.08 | 831.10 |
| 5 | 282.57 | 118.86 | 190.11 | 87.82 | 156.66 | 98.97 | 11,113.801 | 21,505.00 | 10,391.20 | 894.89 |
| 6 | 268.65 | 134.94 | 188.96 | 131.20 | 133.12 | 106.13 | 11,506.8602 | 22,293.45 | 10,786.59 | 903.10 |
| 7 | 282.25 | 110.86 | 200.46 | 116.44 | 161.67 | 117.31 | 11,814.661 | 24,329.40 | 12,514.74 | 943.67 |
| 8 | 318.19 | 126.99 | 194.56 | 113.38 | 162.78 | 107.09 | 12,181.5529 | 25,830.73 | 13,649.18 | 1047.15 |
| 9 | 324.15 | 164.39 | 226.84 | 118.69 | 198.86 | 93.06 | 13,471.5765 | 27,936.05 | 14,464.47 | 1239.81 |
| 10 | 344.50 | 155.59 | 211.73 | 122.96 | 196.45 | 118.77 | 13,786.5861 | 30,475.00 | 16,688.41 | 1265.30 |
| 11 | 350.92 | 183.98 | 240.31 | 139.40 | 167.58 | 118.81 | 14,416.1753 | 32,607.14 | 18,190.97 | 1395.27 |
| 12 | 386.77 | 178.95 | 231.68 | 148.91 | 169.39 | 119.30 | 14,838.8277 | 37,420.49 | 22,581.67 | 1501.13 |
| 13 | 358.79 | 159.28 | 228.62 | 149.05 | 174.61 | 119.65 | 14,275.8021 | 35,342.99 | 21,067.19 | 1368.31 |
| 14 | 379.97 | 165.36 | 249.43 | 146.40 | 190.98 | 118.86 | 15,041.8218 | 35,778.61 | 20,736.78 | 1522.54 |
| 15 | 387.37 | 178.49 | 247.30 | 148.43 | 182.26 | 119.15 | 15,195.1235 | 31,890.75 | 16,695.63 | 1558.50 |
| 16 | 364.96 | 197.62 | 245.53 | 132.81 | 189.51 | 119.58 | 15,050.0093 | 31,000.00 | 15,949.99 | 1499.40 |
| 17 | 358.21 | 169.93 | 256.18 | 129.27 | 189.03 | 118.39 | 14,658.0997 | 28,754.55 | 14,096.45 | 1450.01 |
| 18 | 360.92 | 172.60 | 225.78 | 148.92 | 174.26 | 119.51 | 14,434.0764 | 27,465.70 | 13,031.62 | 1390.41 |
| 19 | 322.15 | 154.47 | 229.83 | 133.87 | 199.14 | 119.54 | 13,920.6668 | 26,714.95 | 12,794.28 | 1255.61 |
| 20 | 335.55 | 147.30 | 209.25 | 108.47 | 172.87 | 118.56 | 13,035.613 | 25,389.00 | 12,353.38 | 1174.63 |
| 21 | 296.40 | 167.11 | 206.24 | 121.43 | 142.45 | 89.38 | 12,187.7462 | 24,040.49 | 11,852.75 | 1055.36 |
| 22 | 275.73 | 153.09 | 199.98 | 95.28 | 164.90 | 95.01 | 11,732.3309 | 22,887.84 | 11,155.51 | 959.52 |
| 23 | 267.59 | 160.44 | 225.76 | 94.03 | 156.50 | 70.68 | 11,596.8066 | 21,937.49 | 10,340.69 | 996.03 |
| 24 | 295.98 | 99.08 | 185.82 | 115.74 | 144.14 | 119.23 | 11,447.352 | 21,312.00 | 9864.65 | 921.21 |
| **Total** | **7650.70** | **3631.54** | **5124.51** | **2951.15** | **3957.45** | **2656.65** | **310,848.56** | **639,357.25** | **328,508.69** | **27,878.43** |

**Table 7.** Optimal generation schedule, cost, selling price and profit for Test Case 4 (Best Compromise Solution).

| A. Optimal Generation Schedule for Test Case 4 (Best Compromise Solution) | | | | | | | | | |
|------|---------|---------|---------|---------|---------|---------|--------|----------|----------|
| Hour | P1 (MW) | P2 (MW) | P3 (MW) | P4 (MW) | P5 (MW) | P6 (MW) | TS (MW) | Emission (Kg) | WS (MW) | PV Share (MW) |
| 1 | 326.45 | 101.35 | 186.46 | 98.17 | 150.86 | 91.71 | 955.00 | 991.51 | 0.00 | 0.00 |
| 2 | 303.95 | 107.68 | 198.92 | 101.68 | 128.74 | 101.03 | 942.00 | 947.51 | 0.00 | 0.00 |
| 3 | 281.69 | 119.22 | 232.89 | 81.07 | 143.44 | 94.68 | 953.00 | 975.98 | 0.00 | 0.00 |
| 4 | 258.20 | 135.39 | 186.99 | 102.72 | 149.44 | 88.95 | 921.69 | 849.56 | 8.31 | 0.00 |
| 5 | 268.76 | 145.56 | 193.86 | 86.33 | 139.38 | 80.16 | 914.06 | 882.95 | 20.31 | 2.63 |
| 6 | 292.74 | 121.33 | 177.03 | 93.73 | 132.47 | 101.55 | 918.84 | 885.64 | 32.31 | 13.85 |
| 7 | 291.77 | 131.62 | 183.65 | 96.06 | 121.16 | 90.69 | 914.93 | 900.13 | 44.31 | 30.76 |
| 8 | 298.97 | 120.68 | 193.18 | 90.61 | 119.46 | 80.31 | 903.21 | 916.65 | 56.31 | 63.48 |
| 9 | 324.98 | 125.01 | 202.09 | 118.72 | 136.55 | 88.13 | 995.48 | 1056.59 | 68.31 | 62.22 |
| 10 | 327.58 | 135.25 | 204.20 | 106.08 | 142.04 | 61.42 | 976.57 | 1070.75 | 80.31 | 93.12 |
| 11 | 352.28 | 112.17 | 217.72 | 103.03 | 122.32 | 74.72 | 982.24 | 1134.22 | 92.31 | 120.00 |
| 12 | 347.07 | 122.85 | 186.34 | 108.57 | 144.80 | 89.03 | 998.66 | 1086.44 | 104.31 | 120.00 |
| 13 | 314.33 | 127.82 | 176.08 | 116.53 | 143.55 | 76.50 | 954.81 | 975.73 | 116.31 | 118.88 |
| 14 | 335.61 | 146.32 | 213.07 | 112.85 | 143.72 | 79.95 | 1031.51 | 1143.19 | 120.00 | 99.49 |
| 15 | 322.65 | 155.12 | 241.44 | 96.25 | 142.49 | 99.82 | 1057.76 | 1180.47 | 120.00 | 85.24 |
| 16 | 334.44 | 157.54 | 226.78 | 108.63 | 139.08 | 86.81 | 1053.29 | 1185.25 | 120.00 | 76.71 |
| 17 | 342.95 | 162.21 | 186.79 | 125.49 | 159.30 | 78.16 | 1054.91 | 1161.91 | 120.00 | 56.09 |
| 18 | 351.52 | 163.81 | 180.67 | 101.87 | 156.07 | 102.82 | 1056.77 | 1161.48 | 120.00 | 26.23 |
| 19 | 306.51 | 160.14 | 194.76 | 120.31 | 150.81 | 101.95 | 1034.48 | 1057.68 | 120.00 | 6.52 |
| 20 | 331.46 | 140.31 | 196.15 | 92.82 | 129.68 | 81.58 | 972.00 | 1055.65 | 120.00 | 0.00 |
| 21 | 351.75 | 122.82 | 222.79 | 98.26 | 135.87 | 91.50 | 1023.00 | 1168.49 | 0.00 | 0.00 |
| 22 | 327.44 | 136.26 | 204.65 | 92.46 | 144.33 | 78.86 | 984.00 | 1066.00 | 0.00 | 0.00 |
| 23 | 302.88 | 129.99 | 188.70 | 97.67 | 170.58 | 85.18 | 975.00 | 980.05 | 0.00 | 0.00 |
| 24 | 265.65 | 140.16 | 211.95 | 95.83 | 154.01 | 92.41 | 960.00 | 929.40 | 0.00 | 0.00 |
| **Total** | **7561.66** | **3220.61** | **4807.17** | **2445.70** | **3400.14** | **2097.93** | **23,533.20** | **24,763.26** | **1463.08** | **975.72** |

**Table 7.** *Cont.*

| B. Cost, Selling Price and Profit for Test Case 4 (Best Compromise Solution) | | | | | |
|---|---|---|---|---|---|
| Hour | Th Cost ($) | WC ($) | PV Cost ($) | TC ($) | SP ($) | Profit ($) |
| 1 | 11,313.10 | 26.59 | 0.00 | 11,339.69 | 21,630.75 | 10,291.06 |
| 2 | 11,163.59 | 26.59 | 0.00 | 11,190.18 | 20,724.00 | 9533.82 |
| 3 | 11,300.92 | 26.59 | 0.00 | 11,327.51 | 21,537.80 | 10,210.30 |
| 4 | 10,982.15 | 30.71 | 0.00 | 11,012.86 | 20,553.00 | 9540.14 |
| 5 | 10,848.76 | 55.33 | 7.55 | 10,911.64 | 21,504.92 | 10,593.28 |
| 6 | 10,907.79 | 102.07 | 141.28 | 11,151.14 | 22,293.51 | 11,142.37 |
| 7 | 10,840.90 | 157.43 | 354.88 | 11,353.21 | 24,329.42 | 12,976.21 |
| 8 | 10,666.11 | 214.29 | 757.03 | 11,637.43 | 25,830.68 | 14,193.25 |
| 9 | 11,798.47 | 271.29 | 741.92 | 12,811.68 | 27,936.16 | 15,124.48 |
| 10 | 11,535.66 | 328.29 | 1110.51 | 12,974.46 | 30,474.88 | 17,500.42 |
| 11 | 11,585.74 | 385.29 | 1507.91 | 13,478.94 | 32,607.17 | 19,128.22 |
| 12 | 11,834.05 | 442.29 | 1574.49 | 13,850.83 | 37,420.41 | 23,569.58 |
| 13 | 11,321.05 | 499.29 | 1417.69 | 13,238.03 | 35,342.89 | 22,104.87 |
| 14 | 12,224.89 | 516.83 | 1186.46 | 13,928.18 | 35,778.49 | 21,850.31 |
| 15 | 12,566.34 | 516.83 | 1016.51 | 14,099.68 | 31,890.71 | 17,791.02 |
| 16 | 12,496.84 | 516.83 | 914.82 | 13,928.49 | 30,999.90 | 17,071.41 |
| 17 | 12,556.04 | 516.83 | 549.59 | 13,622.46 | 28,754.62 | 15,132.16 |
| 18 | 12,588.57 | 516.83 | 300.88 | 13,406.28 | 27,465.67 | 14,059.40 |
| 19 | 12,338.22 | 516.83 | 53.85 | 12,908.90 | 26,715.04 | 13,806.14 |
| 20 | 11,488.50 | 516.83 | 0.00 | 12,005.33 | 25,389.00 | 13,383.67 |
| 21 | 12,101.88 | 26.59 | 0.00 | 12,128.47 | 24,040.50 | 11,912.03 |
| 22 | 11,634.79 | 26.59 | 0.00 | 11,661.38 | 22,887.84 | 11,226.46 |
| 23 | 11,583.47 | 26.59 | 0.00 | 11,610.06 | 21,937.50 | 10,327.44 |
| 24 | 11,428.16 | 26.59 | 0.00 | 11,454.75 | 21,312.00 | 9857.25 |
| Total | 279,105.98 | 6290.22 | 11,635.37 | 297,031.57 | 639,357.25 | 342,325.68 |

*4.7. Analysis and Discussion*

The EO algorithm is employed to analyze the optimal generation schedule for a hybrid thermal-solar PV-wind system under the deregulated environment with the objective is to maximize the profit of the operator and to minimize emission content for the given power demand and tariff. The cost due to uncertainty of RER in meeting the load demand is also included in the model. The effect of integrating solar and wind power units is studied under (i) profit maximization, (ii) emission minimization and (iii) profit-emission optimization.

According to the results mentioned in Sections 4.4–4.6, it is observed that profit increases when more and more renewable units are added to the thermal system. On the other hand, emission content becomes reduced with the addition of more renewable units. For the multiobjective profit-emission optimization case, the improvement in both profit and emission can be seen to lie between the conditions (i) and (ii).

**5. Conclusions**

In this paper Equilibrium Optimization (EO) is applied for the solution of the optimal generation schedule problem of a hybrid thermal-solar-wind test system such that the profit is maximized and the pollution content becomes reduced. The practical constraints

of non-convexity, non-linearity associated with the thermal unit, probabilistic terms due to wind and solar system are included in the cost function and analyzed under dynamic conditions. The performance of EO is also compared and validated with well-known PSO and ABC algorithms.

The simulation results indicate that:

- The EO is not significantly dependent on algorithm-specific control parameters. The results are found to vary in a very narrow band with variations in control parameters.
- As the population size increases, EO gives more promising results. However, an increase in population size leads to increased computational time too.
- EO has a unique embedded mechanism for exploration and exploitation which leads to the global best solution.
- The increase in profit and decrease in emission are computed for the integration of solar, wind and wind-solar units in the existing thermal power generation system.
- It is verified that the higher the integration of RER, the greater is the profit, even after including the uncertainty costs of the renewable energy in the model.
- EO is found to produce well-distributed Pareto-optimal solutions for the multiobjective problem. For all the tested cases it is observed that EO is capable of dealing with complex operational constraints under the dynamic environment in an efficient manner.
- The proposed work is beneficial for designing hybrid renewable power systems with optimal capacities for given conditions to achieve desired profit and to reduce emission.

**Author Contributions:** Conceptualization, S.M.D. and H.M.D.; methodology, M.P.; software, S.R.S.; validation, S.M.D., H.M.D. and M.P.; formal analysis, S.R.S.; investigation, S.M.D.; resources, H.M.D.; data curation, M.P.; writing—original draft preparation, S.M.D. and H.M.D.; writing—review and editing, H.M.D., M.P. and S.R.S.; visualization, S.M.D.; supervision, M.P. and S.R.S.; project administration, S.M.D. and H.M.D.; funding acquisition, M.P. and S.R.S. All authors have read and agreed to the published version of the manuscript.

**Funding:** Woosong University's Academic Research Funding—2021.

**Institutional Review Board Statement:** Not applicable.

**Informed Consent Statement:** Not applicable.

**Data Availability Statement:** Not applicable.

**Acknowledgments:** The authors acknowledge the support provided by Woosong University's Academic Research Funding—2021.

**Conflicts of Interest:** The authors declare no conflict of interest.

## Appendix A

**Table A1.** Cost and emission coefficients and generation limits of thermal units, power demand and respective market selling price.

| Unit | $a_i$($/MW$^2$h) | $b_i$($/MWh) | $c_i$($/h) | Pmin (MW) | Pmax (MW) | $\alpha_i$(Kg/MW$^2$h) | $\beta_i$(Kg/MWh) | $\gamma_i$(Kg/h) | UR (MW/h) | DR (MW/h) |
|---|---|---|---|---|---|---|---|---|---|---|
| 1 | 0.007 | 7 | 240 | 100 | 500 | 0.00419 | 0.32767 | 13.8593 | 80 | 120 |
| 2 | 0.0095 | 10 | 200 | 50 | 200 | 0.00419 | 0.32767 | 13.8593 | 50 | 90 |
| 3 | 0.009 | 8 | 220 | 80 | 300 | 0.00683 | −0.54551 | 40.2669 | 65 | 100 |
| 4 | 0.009 | 11 | 200 | 50 | 150 | 0.00683 | −0.54551 | 40.2669 | 50 | 90 |
| 5 | 0.008 | 10.5 | 220 | 50 | 200 | 0.00461 | −0.51116 | 42.8955 | 50 | 90 |
| 6 | 0.0075 | 12 | 190 | 50 | 120 | 0.00461 | −0.51116 | 42.8955 | 50 | 90 |

| Hour | 1 | 2 | 3 | 4 | 5 | 6 | 7 | 8 | 9 | 10 | 11 | 12 |
|---|---|---|---|---|---|---|---|---|---|---|---|---|
| PD (MW) | 955 | 942 | 953 | 930 | 935 | 963 | 989 | 1023 | 1126 | 1150 | 1201 | 1235 |
| SP ($/MW) | 22.65 | 22 | 22.6 | 22.1 | 23 | 23.15 | 24.6 | 25.25 | 24.81 | 26.5 | 27.15 | 30.3 |

| Hour | 13 | 14 | 15 | 16 | 17 | 18 | 19 | 20 | 21 | 22 | 23 | 24 |
|---|---|---|---|---|---|---|---|---|---|---|---|---|
| PD (MW) | 1190 | 1251 | 1263 | 1250 | 1221 | 1202 | 1159 | 1092 | 1023 | 984 | 975 | 960 |
| SP ($/MW) | 29.7 | 28.6 | 25.25 | 24.8 | 23.55 | 22.85 | 23.05 | 23.25 | 23.5 | 23.26 | 22.5 | 22.2 |

**Table A2.** Data for Solar PV units and wind farm.

| Type of System | No. of Units | Rated Power (MW/Unit) | DC ($/MWh) | $kp$ | $kr$ | $k$ | $c$ | $Vci$ (m/s$^2$) | $Vr$ (m/s$^2$) | $Vco$ (m/s$^2$) |
|---|---|---|---|---|---|---|---|---|---|---|
| Solar PV | 2 | 60 | 12 | 1.5 | 3 | - | - | - | - | - |
| Wind | 2 | 60 | 1.75 | | | 2 | 10 | 3 | 16 | 25 |

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
