# Peer review of "Multiobjective Scheduling of Hybrid Renewable Energy System Using Equilibrium Optimization"

_energies, doi:10.3390/en14196376_

Round 1

Reviewer 1 Report

It seems the authors would like to apply so-called “Equilibrium Optimization” to any of the available systems. How the reported results can be interesting and useful for the readers?

Reviewer 2 Report

This research implemented a novel optimization algorithm in the hybrid energy system scheduling problem. The optimization problem was modeled. The profit maximization, emission minimization, and simultaneous optimization of both were defined three optimization objectives. The optimization process was well established. Sufficient analysis and comparisons were presented. This research fit the journal scope very well, and is of interests and topical to the audience of the journal. Before acceptance, the authors should consider the following comments.

1 The abstract section should describe what you have done and the main achievements you have reached in this research. Current abstract introduces why to conduct this research. Please add some main findings in you abstract.

  1. Some of the abbreviation explanation is missing. For instances, page 2 line 47: TVAC, EMA.

  1. Typos, Page 3 Line 114, “Were” should be “where”.

  1. In page 5 line 170, the minimum of reciprocal of profit and total emissions formed the third optimization objective. This simple combination of two items should be further explained, since the combination is less defined.

  1. In page 6 line 195, the equation for multi-objective ranking is incorrect. It should be (Fmax-Fr)/(Fmax-Fmin).

  1. In terms of the multi objective of profit and emission, there is a way to convert the emission to profit, since there is carbon trade market. China launched carbon trade market in July 2021. The European Union Emissions Trading System set up in 2005, is an accuracy, efficacy way to measure the carbon emissions. Those market provide an effective way to convert the emissions to cost if the company should purchase carbon credits, or the profit if the company could sell carbon credits. This is an alternative for the optimization of simultaneous.

  1. In the modeling techniques, from Line 108 to Line 166. The generation cost of thermal is defined as a fitting curve of the amount of thermal generation (equation 4), as well as the total emission (equation 15). However, the modeling error/accuracy of those curve fittings remains unknown. Please clarify the modeling error.

Reviewer 3 Report

The authors desribe the application of Equilibrium Optimisation (EO) to dispatch optimisations. They choose multiobjctive optimisation in the two dimensions revenue and emissions as an example, while they include uncertainties of renewable electricity production as additional costs. Their study supports the claim that EO offers more computational performance when compared to particle swarm optimisation and artificial bee colony.

The paper makes an overall sound and adequate impression. However, formating shoud be improved. I.e. paragraph headings are sometimes at the last line of a page and small number of lines before or after figures are sometimes easily overlooked. (Figures might be better at the very top or bottom of the pages.)

General Content:

  1. It ramains unclear, what the "best compromise" between cost and emissions should be. This would require some weighting which of the objectives is more or less important. I.e., what is the "best solution" in line 325? Later, even a pareto front is shown, which implies that no weighting happened.
  2. Is there any reason why no quadratic minimisation is used for the single objective case? This should be possible and guarantees an optimal result, as it does not rely on heuristics.

Detailled aspects of the content:

  1. Due to the black bars in Figure 1, it looks as if the ISO only consideres the sums of both, supply and demand. This is not the case.
  2. In Eq. (1), "RV - TC" is not the maximum.
  3. From Eq. (2) and (3) it an be guessed that 24 time steps are considered in the example. This should be made explicit. (Also, formating an description of these equations are rather hard to follow.)
  4. In lines 333f, information about software and comuter is irrelevant (or lacks precision, if really needed).
